# Two-Window Approach to Monitor and Assess Cellular and Humoral Immune Responses in Poultry

**Gisela F. Erf** *[ID], **Hyeonmin R. Kong, Daniel M. Falcon** [†] **and Kristen A. Byrne** [‡][ID]

Department of Poultry Science, Center of Excellence for Poultry Science, University of Arkansas System Division of Agriculture, Fayetteville, AR 72701, USA

* Correspondence: gferf@uark.edu; Tel.: +1-479-575-8664

† Current address: Division of Molecular Medicine, University of New Mexico, Albuquerque, NM 87131, USA.

‡ Current address: National Animal Disease Center, USDA-ARS, Food Safety Enteric Pathogens Research, Ames, IA 50010, USA.

**Abstract:** As previously reported, inflammatory activity initiated by intradermal injection of multiple growing feather (GF)-pulps of a chicken with lipopolysaccharide, and the subsequent periodic sampling of GFs and blood, enables the longitudinal evaluation of in vivo tissue- and systemic-inflammatory activities by ex vivo laboratory analyses. To demonstrate the suitability of this two-window approach to monitor and assess vaccine responses, two groups of chickens were immunized by intramuscular injection of mouse IgG (mIgG), mIgG in alum adjuvant (Alum&mIgG), or PBS-vehicle (Group I and II at 7- and 7- and 11-weeks, respectively). Plasma levels of mIgG-specific antibodies were monitored by ELISA for 28 days post-primary- and secondary-immunizations. To examine the cellular responses, 20 GF-pulps per bird were injected with mIgG on Day-10 or Day-5 post-primary- or -secondary-immunization, respectively. Two GFs were collected before- and at various times (0.25 to 7 days) post-injection for leukocyte population- and cytokine mRNA expression-analyses. The observed primary- and secondary-antibody response profiles were as expected for a T-dependent antigen. Leukocyte- and cytokine-profiles established in GF-pulps revealed temporal, qualitative, and quantitative differences in local naïve, primary, and secondary leukocyte-effector responses to antigen. This study demonstrates the unique opportunity in the avian model to monitor both cell- and antibody-mediated immune responses using minimally invasive techniques.

**Keywords:** humoral immunity; leukocytes; cell-mediated immunity; T cell subsets; B cells; cytokines; chicken; skin bioassay; adaptive immune response; adjuvant

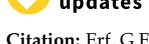



## 1. Introduction

In animal studies of immune system responses, soluble indicators of immune system activities, such as antibodies, cytokines, and acute phase proteins can be detected in blood and other tissue fluids that can be sampled using minimally invasive techniques. Of these, measurements of antigen-specific antibody levels are particularly informative regarding the temporal, qualitative, and quantitative aspects of the humoral immune response to the antigen. On the other hand, cellular immune system responses and activities are difficult to study and monitor because most of the direct interactions between antigen and immune system components occur in complex tissues and sampling of affected tissues for ex vivo analysis requires invasive, and often terminal procedures. Moreover, in the case of most T cells, their activation is restricted and controlled by the specificity of their antigen-receptor for the processed antigen-peptide presented in association with self-MHC molecules on the surface of cells. This need for MHC-match constitutes a particular challenge for the study of T cell responses in vivo and in vitro, especially when examining activities in outbred populations [1].

The skin and its derivatives have been used extensively as a test-site and indicator of cellular immune responses in mammals and birds [1–10]. For the skin test, antigen is

injected intra-dermally, and the time, type, and extent of a visible skin reaction is monitored. The nature (e.g., induration and erythema at 24–48 h) and extent (e.g., increase in skin thickness) of the response to the injected antigen serve as indicators of whether or not an individual has previously been exposed to the antigen and, if so, the kind and relative strength of the immune response that was generated to the antigen. However, to gain insight into the cellular and molecular processes underlying the visible response to antigen, skin biopsies need to be conducted for ex vivo sample analysis, limiting observations to snap shots of the response at different times in different individuals.

Past research in our laboratory provided proof of concept that the growing feather (**GF**) in chickens is a suitable skin test-site to monitor local in vivo tissue/cellular responses in an individual [10,11]. In chickens, the living portion (pulp) of a GF is a column of approximately 8–10 mm in height with a 2–3 mm diameter [10]. The pulp of the GF consists mostly of inner dermis surrounded by epidermis and an outer sheath [12]. Intradermal (**i.d.**) micro-injection of the pulp of several GFs of a chicken with test-material [e.g., (recall-) antigens, adjuvants, immunomodulators, nanomaterials, etc.] and collection of individual injected-GFs at various times post-injection (minutes, hours, days) for ex vivo analysis, enables the monitoring of local in vivo immune activities to test materials in the same individual. Unlike other cutaneous test-sites, injection and collection of GFs are minimally invasive procedures, the local tissue response is contained in the GF sample, and each GF sample constitutes a defined biopsy unit (in vivo test-tube) that provides a rich source of cells, RNA, DNA, and proteins for ex vivo analyses [3,10,13–16]. Hence, using the GF in vivo test-tube, we have the ability to examine temporal, qualitative, and quantitative aspects of an individual's cellular/tissue responses to test-material in a complex tissue.

In a recent study, we successfully combined the vivo test-tube system with sampling of the blood to monitor the local- and systemic-inflammatory responses following i.d. GF-pulp injection of lipopolysaccharide in broilers [17]. The objective of this study was to demonstrate the application of this two-window approach to simultaneously examine and assess adaptive cellular and humoral immune responses, over-time, in an individual. Specifically, the test-antigen chosen was mouse IgG, a T-dependent xenogeneic protein-antigen, known to stimulate antibody production in chickens. Two mIgG immunization treatments were tested: mouse IgG in PBS (**mIgG**) and mIgG in 15% alum adjuvant (**Alum&mIgG**) [18]. For primary and secondary immunizations, chickens were injected into the breast muscle with PBS-vehicle (unsensitized chickens), mIgG, or Alum&mIgG. The primary and secondary IgM- and IgG-antibody responses to mIgG-antigen were monitored in plasma over 4 weeks each. To examine the local cellular effector responses, GF-pulps of immunized chickens were injected with mIgG-antigen during the height of the primary or secondary responses and the local cellular responses monitored over the course of 5 to 7 days. The mIgG-antigen also was injected into GFs of age-matched, unsensitized chickens to examine the local cellular responses during a first encounter with test-antigen.

As expected, the GF in vivo test-tube system successfully provided novel insight into temporal, qualitative, and quantitative aspects of the local leukocyte responses (infiltration and cytokine expression) to protein antigen in a complex tissue, while blood sampling revealed classic primary and memory antibody response profiles to the T-dependent antigen.

## 2. Materials and Methods

### 2.1. Experimental Animals

Males from the Light-brown Leghorn line of chickens maintained by G. F. Erf at the University of Arkansas System Division of Agricultural (UADA) Poultry Research Farm in Fayetteville, AR, USA were used for these studies. Chicks were hatched, tagged, not vaccinated, and placed in floor pens on wood shavings litter in a HEPA-filtered room in the UADA Poultry Health Laboratory in Fayetteville, AR. All studies were conducted with the approval of the University of Arkansas Institutional Animal Care and Use Committee (IACUC).

## 2.2. Immunization Protocols

At 7 weeks of age, the chickens were randomly assigned to three immunization treatments and split into two groups of 12 birds each. Group I was subjected to one intramuscular (i.m.) immunization at 7 weeks of age and Group II was subjected to two i.m. immunizations with the same treatment administered at 7- and again at 11-weeks of age. Immunization treatments consisted of vehicle (**PBS**; endotoxin-free Dulbeccos's phosphate buffered saline, Sigma-Aldrich, St. Louis, MO, USA) injection (0.1 mL) or 26 μg of mouse IgG (mIgG)-antigen (Rockland, Inc., Gilbertsville, PA, USA) administered in a 0.1 mL volume in PBS (**mIgG**) or mIgG-antigen mixed with 15% alum adjuvant(Alhydrogel; InvivoGen, San Diego, CA, USA) in PBS (**Alum&mIgG**). All immunization treatments were administered into the left breast muscle for the primary immunization (Group I and II) and the right breast muscle for the second immunization (Group II) using 1 mL syringes with 25 gauge × 12.5 mm needles (Becton, Dickinson, and Company, Franklin Lakes, NJ, USA).

## 2.3. Blood Sampling

To determine the antibody response to mIgG, 0.5–1 mL of blood was collected from the wing vein into 1 mL heparinized syringes before (0) and at 3, 7, 10, 14, 21, and 28 d after the first immunization (Group I and II). For Group II, blood samples were also collected before (0 d; Day 28 of primary) and at 3, 5, 7, 10, 14, 21, and 28 d after the second immunization. Plasma was isolated from the blood samples and stored at $-20\,^\circ$C until determination of mIgG specific chicken-IgM and -IgG antibody levels by ELISA.

## 2.4. Intra-Dermal Injection of Mouse IgG-Antigen into GF-Pulps

Injection of mIgG-antigen into the GF was carried out 10 d after the first immunizations in Group I and 5 d after the second immunization in Group II, during the anticipated height of the primary and secondary responses, respectively. Specifically, the pulp of the 18-day-old regenerating GF was injected with 10 μL of mIgG (1 mg/mL) in PBS; 10 GFs on each the left and the right breast tract [10]. Two GFs were collected before (0) and at 0.25 (6 h), 1, 2, 3, 4, 5, and 7 d post-GF injection from each chicken. At each time-point, one of the GFs was placed in ice-cold PBS for same day cell-population analysis; the other GF was placed in RNAlater® RNA preservation buffer (Ambion, Thermo Fisher Scientific Inc., Waltham, MA, USA) and stored following the manufacturer's instruction until use for RNA isolation and quantitative RT-PCR.

## 2.5. ELISA to Determine the Plasma Levels of Mouse IgG-Specific IgM and IgG Antibodies

Flat-bottom, 96-well ELISA plates were coated overnight at 4 °C with mouse IgG (5 μg/mL; 0.1 mL/well) prepared in 0.05 M carbonate-bicarbonate buffer (pH 9.6) and plates were incubated over night at 4 °C. Following incubation, the plates were washed five times with wash solution (50 mM Tris, 0.14 M NaCl, and 0.05% Tween 20 (TBS-T), pH 8.0). The plate wells were then filled with 200 μL of blocking buffer (TBS 1% BSA, pH 8.0) and incubated at room temperature (**RT**) for 30 min. Following the blocking step, the plates were washed five times and 100 μL/well of diluted plasma samples or standard amounts (156 to 0.312 ng/mL) of mIgG-specific chicken IgG antibody (Invitrogen; Thermo Fisher Scientific, Waltham, MA, USA), prepared in diluent (TBS-T 1% BSA, pH 8.0) were added and incubated at RT for 1 h. All samples were subjected to both IgM- and IgG-ELISAs at the same time and tested at three dilutions, in triplicate wells. Plasma dilutions ranged from 1:200 to 1:320,000, depending on the time post-immunization and antibody isotype tested. Dilutions were prepared empirically to fall within the limits of the standard curve. Because chicken IgM antibodies to mIgG were not available, the IgG standards were also included in each IgM-ELISA plate as positive controls, and to provide a consistent method to determine the relative amounts of mIgG-specific IgM antibodies. Additional controls included a "blank" to determine the background color and a non-specific binding control. Following incubation, the plates were washed as before, 100 μL HRP-conjugated goat-anti-chicken IgG- or IgM-detection antibody (Bethyl Laboratories Inc., Montgomery,

TX, USA) was added (1:10,000 dilution), and the plates were incubated for 1 h at RT. Following the incubation, the plates were washed five times and 100 μL of substrate (TMB One Component Microwell Substrate, Bethyl Laboratories Inc.) was added to each well. The plates were incubated for 15 min at RT before 100 μL of stop solution (2M $H_2SO_4$) was added to all wells and absorbance was determined with an ELISA plate spectrophotometer at a 450 nm wavelength. Using JMP Pro 10 Statistical Software (SAS Institute Inc., Cary, NC, USA), a standard curve equation was generated using a 4-parameter logistic equation.

### 2.6. Preparation and Immunofluorescent Staining of Pulp Cell Suspensions for the Leukocyte Population Analysis by Flow Cytometry

Pulp cell suspensions of the collected GFs were prepared and immunofluorescently stained using a two- and three-color direct staining method, as described [10,17]. Fluorescence-conjugated mouse-anti-chicken (mac) leukocyte specific monoclonal antibodies (all IgG1 isotype) were purchased from Southern Biotechnology Associates, Inc., Birmingham, AL and included: CD45 conjugated with SPRD (CD45-SPRD; leukocytes), CD4-FITC (T helper cells), CD8α-PE (cytotoxic lymphocytes), Bu-1-FITC (B cells), IgM-PE, γδ T cell receptor (TCR)-PE (γδ T cells), and KUL01-FITC (macrophages). Staining controls, compensation procedures, and acquisition set-up for flow cytometric analysis were as described in [10,17]. For each sample, forward scatter (FSC), side scatter (SSC), FITC-, PE- and SPRD-fluorescence data based on 10,000 cells were acquired using a BD FACSsort and CellQuest software (BD Immunocytometry Systems, San Jose, CA, USA). All data were analyzed using FlowJo software (FlowJo, LLC, Ashland, OR, USA). To compare samples on a relative quantitative basis, data for each cell type examined were expressed as the percentage of total pulp cells in the suspension (% pulp cells). Because antibodies specific for chicken heterophils (avian counterpart of neutrophils) were not available, heterophil populations in the pulp cell suspension were estimated based on size (FSC) and granularity (SSC) characteristics of CD45+ leukocytes, as described [19].

### 2.7. RNA Isolation, Quantification, and cDNA Synthesis

For RNA isolation, GF pulps which were stored in RNAlater® at −20 °C were homogenized with Tissue Tearor™ (BioSpec Products, Inc., Bartlesville, OK, USA, Model: 985370-395) in TRIzol® provided with the Direct-Zol RNA Kits (Zymo Research, Irvine, CA, USA) and the total RNA was isolated from homogenates using the same kit, with in-column DNaseI digestion. Total RNA (1.0 μg/sample) was transcribed to cDNA using a high capacity cDNA reverse transcription kit, according to the manufacturer's protocol (Applied Biosystems, Foster City, CA) and [17,20].

### 2.8. Relative Expression of Cytokines

Intron-spanning primers and probes for the target genes used in this study were as described [13,17]. Real-time PCR was performed according to [17,20]. A pool of cDNA from non-injected GF pulps was used as the calibrator sample. The relative gene expression was determined by the delta delta Ct (ΔΔCt) method [21] and data were expressed as the fold change relative to the calibrator sample.

### 2.9. Statistical Analysis

The experimental unit was the individual chicken with four chickens per treatment (PBS, mIgG, Alum&mIgG) within a group. Using Sigma Plot 14.5 Software (Systat Software, Inc., San Jose, CA 95110, USA), one- and two-way analysis of variance (ANOVA) was conducted for the GF-pulp data, and two-way repeated-measures (RM) ANOVA for the antibody data, to determine the main effects of time, treatment, and treatment by time interactions for each aspect examined. In the absence of time by treatment interactions, (RM)-ANOVA was followed by Holm–Sidak multiple means comparison tests on the main effect means. In the presence of time by treatment interactions, and for vehicle control samples, the effect of time was determined by one-way (RM)-ANOVA for each treatment separately and treatment comparisons were made at each time point by one-way

ANOVA. Significant one-way (RM)-ANOVA were followed by Holm–Sidak multiple mean comparisons. For gene-expression mRNA data, one- and two-way ANOVA were carried out on ddCt data. Differences were considered significant at $p \leq 0.05$ for all analyses.

## 3. Results

### 3.1. Humoral Response to mIgG

In this time-course study, mIgG-specific IgM and IgG antibody levels were monitored in plasma over 28 d, each following a primary and secondary i.m. immunization with PBS-vehicle, mIgG antigen in PBS (mIgG) or mIgG antigen in 15% alum adjuvant in PBS (Alum&mIgG).

#### 3.1.1. Antigen-Specific IgM Antibody Responses

Intra-muscular injection with PBS (vehicle control) did not affect the mIgG-specific IgM levels throughout the course of both the primary and secondary responses (Figure 1A,B).

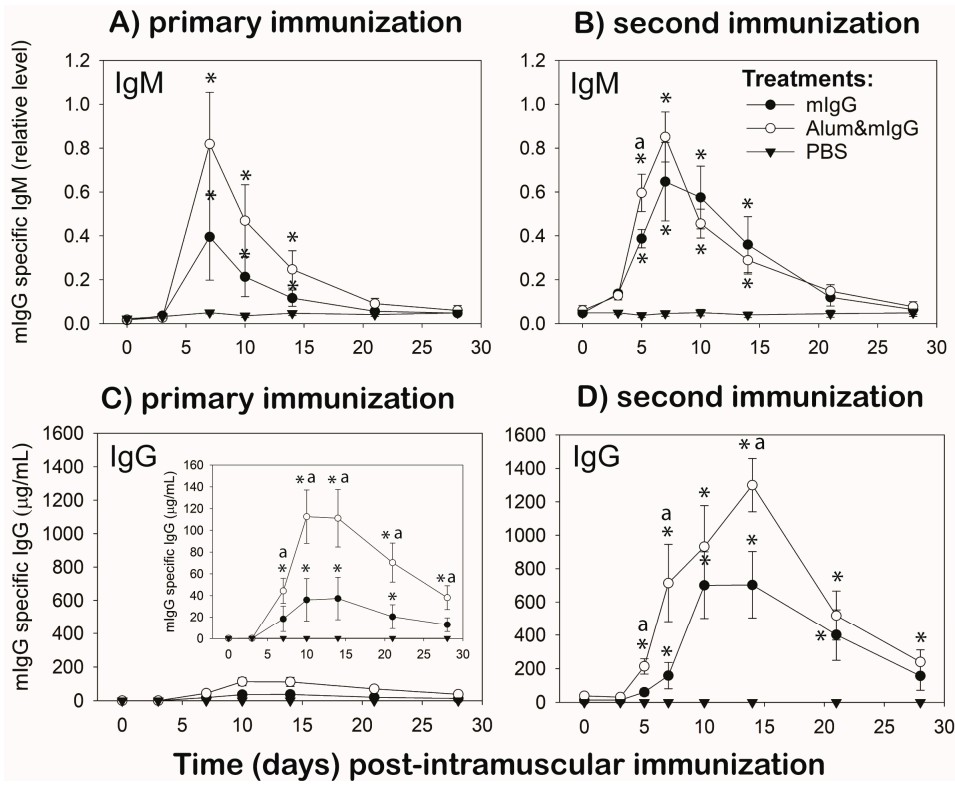

**Figure 1.** Primary and secondary antibody responses in chickens immunized with mouse IgG-antigen administered in PBS or mixed with alum adjuvant. (**A**) primary and (**B**) secondary mIgG-specific IgM antibody response; (**C**) primary and (**D**) secondary mIgG-specific IgG antibody response. Male Light-brown Leghorn chickens were injected i.m. with 26 µg of mouse IgG (mIgG) in endotoxin-free (EF) PBS or mIgG mixed with 15% alum adjuvant (Alum&mIgG) in EF-PBS; four chickens per immunization treatment. Another four chickens were injected i.m. with the same volume (0.1 mL) of EF-PBS vehicle. The immunizations were conducted when the chickens were 7 and 11 weeks of age (1° and 2° vaccination, respectively). Blood samples were collected before and at 3, 7, 10, 14, 21, and 28 d post−1°- and at 3, 5, 7, 10, 14, 21, and 28 post−2°-vaccination. ELISA was used to determine the plasma levels of mIgG-specific IgG (µg/mL) and IgM (relative levels). Data are the mean ± SEM, based on four chickens per treatment. The primary IgG response is shown on the same plasma concentration scale as the secondary response and at a 10-fold lower scale range (insert). * Within a treatment group, mean values at a time-point are higher ($p \leq 0.05$) than before i.d. injection of mIgG antigen (0 d). a: indicates differences between mean values of mIgG and Alum&mIgG vaccination treatment at a time-point ($p \leq 0.05$).

For the primary immunizations, both mIgG immunization treatments (mIgG and Alum&mIgG) resulted in elevated ($p \leq 0.05$) plasma levels of mIgG-antigen specific IgM antibodies at 7, 10, and 14 d post-immunization compared to 0 d (before). Overall, antigen-specific IgM antibody levels were higher in chickens immunized with Alum&mIgG than mIgG, although not significantly at the individual time-points examined. IgM returned to near base-line levels at 21 and 28 d for all immunization treatments (Figure 1A).

A second immunization with the same treatments on 28 d post-primary immunization resulted in elevated antigen-specific IgM antibody levels at 3 to 14 d for both immunization treatments, with peak levels observed at 7 d. IgM levels were similarly high for both immunization treatments, except for higher IgM levels with Alum&mIgG compared to mIgG on 5 d post second immunization (Figure 1B).

Overall, the mIgG-specific IgM response profiles were similar with the primary and secondary immunization, except, following the secondary immunizations there were earlier increases (3 d) in mIgG-specific IgM with both treatments, higher levels with the mIgG immunization treatment, and less variation at each time-point (Figure 1A,B).

### 3.1.2. Antigen-Specific IgG Antibody Responses

Intra-muscular injection with PBS (vehicle control) did not affect mIgG-specific IgG levels throughout the course of both the primary and secondary response (Figure 1C,D).

For primary immunizations, both mIgG and Alum&mIgG immunization treatments resulted in elevated ($p \leq 0.05$) plasma levels (µg/mL) of mIgG-antigen specific IgG antibodies from 7 through 21 d (mIgG) or 28 d (Alum&mIgG) post-immunization compared to 0 d (before), reaching peak-levels on 10 and 14 d. During the primary response, antigen-specific IgG antibody levels were approximately three-fold higher ($p \leq 0.05$) on 10, 14, 21 and 28 d post-immunization in chickens immunized with Alum&mIgG compared to mIgG (Figure 1C).

A second immunization with the same treatments on 28 d post-primary immunization resulted in elevated antigen-specific IgG antibody levels at 5 to 28 d for Alum&mIgG-, and 7 to 21 d for mIgG. Levels of mIgG-specific IgG levels were higher at 5, 7, and 14 d post-secondary immunization with Alum&mIgG compared to mIgG treatment (Figure 1C).

For both treatments, the mIgG-specific IgG response was higher and more rapid following the secondary immunization compared to the primary immunization. Peak IgG levels were approximately 10-fold (120 µg/mL vs. 1200 µg/mL) higher with Alum&mIgG and 15-fold (40 µg/mL vs. 600 µg/mL) higher with mIgG in the primary vs. the secondary response (Figure 1C,D).

Independent of the vaccination treatment, the temporal, qualitative, and quantitative differences in the mIgG-specific IgG antibody response to primary- compared to secondary-immunizations suggest a T-dependent response with isotype switching and memory phenotype (Figure 1).

### 3.2. *Leukocyte Infiltration Profiles in GF-Pulps following Intradermal Injection of mIgG-Antigen in Unsensitized Chickens and Chickens Immunized with mIgG or Alum&mIgG*

In this study, GF-pulps were i.d. injected with mIgG-antigen in PBS-immunized (unsensitized) chickens (Figure 2A) and chickens previously immunized (i.m.) once (Figure 2B) or twice (Figure 2C) with mIgG or Alum&mIgG, to establish the local infiltration response profiles of heterophils, monocyte/macrophages, γδ T cells, CD4+ T cells, CD8+ T cells, and B cells over 5 to 7 d post-GF injection (Figure 2).

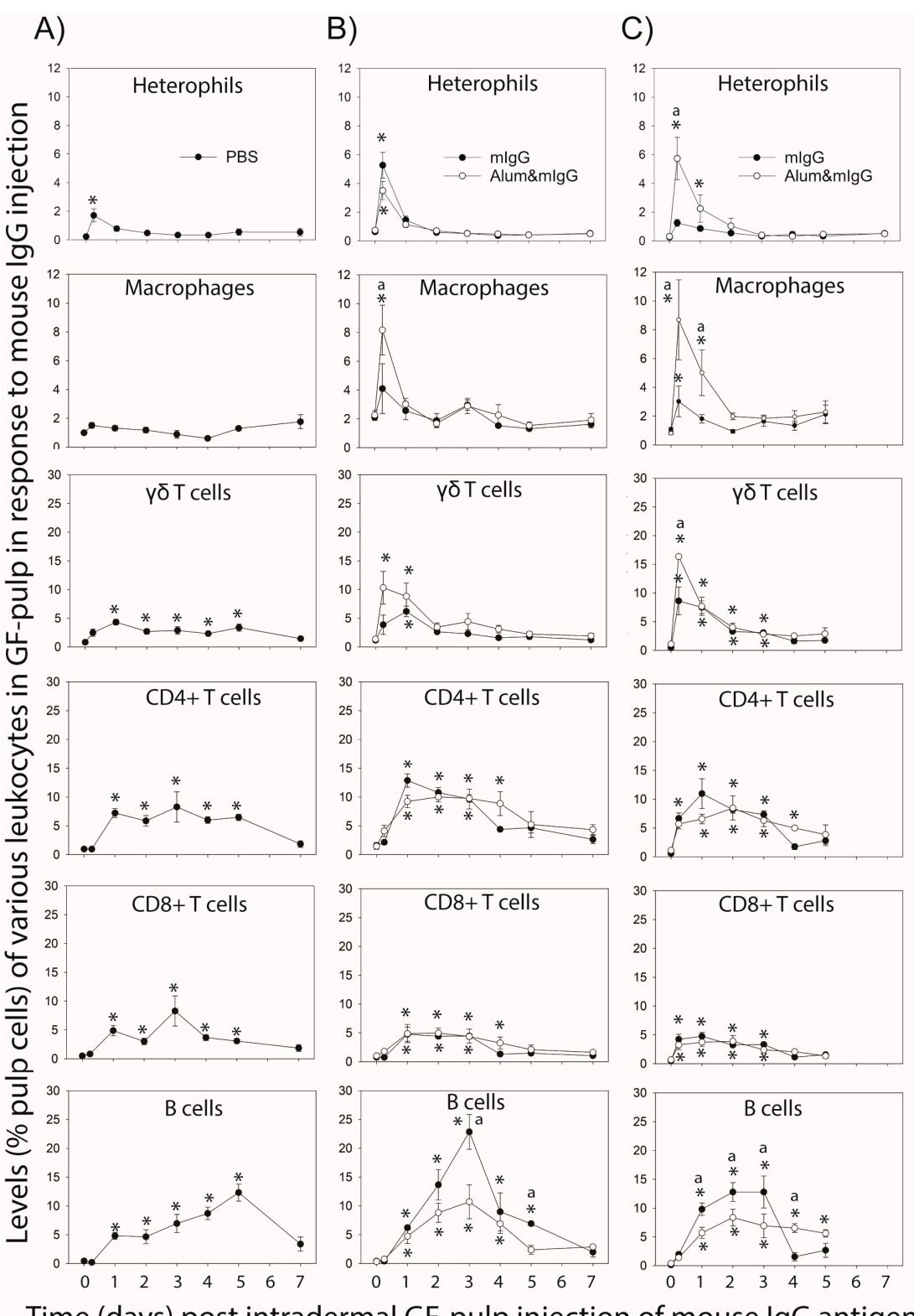

**Figure 2.** Leukocyte population profiles in response to intradermal (i.d.) injection of mouse IgG-antigen into the pulp of growing feathers (GF) in unsensitized (sham-(PBS) immunized) chickens (**A**), and in immunized chickens on Day 10 after the primary (**B**) or Day 5 after the secondary (**C**) intramuscular (i.m.) immunization with mouse IgG in PBS (mIgG) or mixed with alum adjuvant (Alum&mIgG). For immunization, male Light-brown Leghorn chickens were injected i.m. with 26 µg of mIgG in endotoxin-free (EF) PBS or mIgG mixed with 15% alum adjuvant in EF-PBS; four chickens

per immunization treatment. Another four chickens were i.m. injected with the same volume (0.1 mL) of EF-PBS vehicle. The 1° and 2° immunizations were conducted when the chickens were 7 and 11 weeks of age, respectively). The pulps of 20 GFs were i.d. injected with mIgG (1 μg/μL; 10 μL/GF) 10 d after the primary, or 5 d after the secondary i.m. immunization, and in age-matched sham (PBS) immunized chickens. Injected GFs were collected before (0) and at 0.25, 1, 2, 3, 4, 5, and 7 d post GF-injection to determine leukocyte profiles. Pulp cell suspensions were prepared from one GF per time-point and per chicken, and pulp cells immunofluorescently stained with chicken-specific fluorescence-conjugated mouse monoclonal antibodies to identify various leukocyte populations. Cell population analysis was carried out by flow cytometry. Percentages of heterophils were based on size (FSC) and granularity (SSC) characteristics of leukocytes (CD45+). Data shown are percentages of total GF pulp cells; mean ± SEM; n = 4 per time-point and treatment. The legends PBS, mIgG, Alum&mIgG within the graphs indicate i.m. immunization treatments. * Within a treatment group, mean values at a time-point are higher ($p \leq 0.05$) than before i.d. injection of mIgG antigen (0 d). a: indicates differences in mean values between mIgG and Alum&mIgG vaccination treatment at a time-point ($p \leq 0.05$).

### 3.2.1. Heterophils

In unsensitized (PBS immunized) chickens and chickens receiving a first immunization (**1° Vac**), heterophil levels (% pulp cells) were only elevated ($p \leq 0.05$) at 6 h after intradermal (i.d.) pulp injection of mIgG-antigen. In 1° Vac birds, heterophil infiltration was similar for both immunization treatments (mIgG and Alum&mIgG), whereas in birds receiving the secondary vaccination (**2° Vac**), heterophil levels were elevated ($p \leq 0.05$) at 6 h only for Alum&mIgG and were higher ($p \leq 0.05$) at 6 h and 1 d in Alum&mIgG compared to the mIgG immunization treatment. Overall, peak heterophil levels were higher (2% vs. 6%) in immunized compared to unsensitized birds.

### 3.2.2. Macrophages

In unsensitized chickens, macrophage levels did not change significantly over time (main effect of time $p = 0.071$), whereas in immunized birds, there was a main effect of time ($p < 0.001$), with macrophage infiltration reaching peak levels at 6 h for both 1° and 2° Vac and remained elevated longer in birds from the 2° Vac group. Macrophage infiltration was higher in Alum&mIgG than in mIgG (approximately 8 % versus 4%, respectively) in both 1° and 2° Vac groups.

### 3.2.3. γδ T Cells

In unsensitized birds, γδ T cell levels were elevated ($p \leq 0.05$) from 1 to 5 d and returned to pre-injection levels on 7 d. In immunized birds, however, γδ levels increased faster (within 6 h) and to higher levels than in the unsensitized birds, but were only elevated ($p \leq 0.05$) at 6 h and 1 d and decreased thereafter to near pre-injection levels. Peak levels were higher in 2° versus 1° Vac birds.

### 3.2.4. CD4+ T Cells

Infiltration levels for CD4+ cells were similar, independent of immunization status (unsensitized, 1° and 2° Vac) and immunization treatment (mIgG, Alum&mIgG). In unsensitized and 1° Vac birds, CD4+ T cell levels were elevated ($p \leq 0.05$) from 1 to 5 d, whereas in 2° Vac birds, similarly high levels were already achieved at 6 h and remained elevated ($p \leq 0.05$) through 3 d, before decreasing to near pre-injection levels by 5 d.

### 3.2.5. CD8+ T Cells

As for CD4+ T cells, infiltration levels for CD8+ cells were similar independent of the immunization status and immunization treatment, whereby levels were similarly elevated ($p \leq 0.05$) on 1 to 5 d, 1 to 3 d, and 6 h to 3 d, for unsensitized, 1° Vac, and 2° Vac, respectively, before decreasing to near pre-injection levels by 5–7 d.

### 3.2.6. B Cells

In unsensitized birds, i.d. mIgG injection resulted in elevated ($p \leq 0.05$) B cell levels on 1 d and continued to increase to peak levels on 5 d before decreasing to near pre-injection levels on 7 d. In immunized birds, there were treatment by time interactions for both 1° and 2° Vac groups. In 1° Vac chickens, B cell levels increased ($p \leq 0.05$) at a steep rate from 1 d to higher peak levels on 3 d with mIgG compared to Alum&IgG. In Alum&mIgG immunized chickens, B cell levels also peaked on 3 d, but at lower levels than with mIgG immunization. B cell levels were elevated ($p \leq 0.05$) on 1–5 d and 1–4 d with mIgG and Alum&mIgG treatments, respectively. In the 2° Vac group, B cell levels in mIgG immunized birds were elevated ($p \leq 0.05$) at similarly high levels on 1 to 3 d, before sharply dropping to preinjection levels on 4 and 5 d. With Alum&mIgG immunization, however, B cell infiltration levels were elevated at similarly high levels from 1 to 5 days. From 1 to 3 d, B cell levels were higher in mIgG than Alum&mIgG immunized birds, while this trend was reversed on 4 and 5 d.

### 3.3. Cytokine mRNA Expression in GF-Pulps following Intradermal Injection of Mouse IgG-Antigen in Unsensitized Chickens and Chickens Immunized with mIgG or Alum&mIgG

The dermis of GF-pulps was injected with mIgG-antigen in PBS-immunized (unsensitized) chickens (Figure 3A) and chickens immunized once (Figure 3B) or twice (Figure 3C) with mIgG or Alum&mIgG, to establish cytokine mRNA expression profiles for IL1β, IL6, IL8, IL10, IL21, IL4, and IFNγ by targeted qRT-PCR over 5 d post GF-injection. There were no treatment by time interactions for any of the cytokines examined, hence only the main effect data are shown and treatment main effects are indicated when present (Figure 3).

### 3.3.1. IL1β, IL6, and IL8 (CXCL8) Expression

The time-course and relative magnitude (fold expression) of innate, pro-inflammatory cytokines IL1β, IL6, and IL8 were similar in mIgG-antigen injected GF-pulps of unsensitized chickens, reaching peak levels within 6 h post-injection, returning to lower levels within 2 d, fluctuating at this lower level thereafter, and returning to near baseline levels by 5 d. Independent of the immunization treatment, the time-course and magnitude of IL1β, IL6, and IL8 mRNA expression in mIgG-antigen injected GF of immunized chickens were nearly identical in 1° and 2° Vac birds. For IL1β, expression levels peaked at 6 h, dropped to lower levels at 1 d, and remained elevated near this level on 2–5 d in 1° Vac birds, and on 2 and 3 d in 2° Vac birds. Overall, IL1β mRNA expression was higher in Alum&mIgG than mIgG immunized birds, especially for 1° Vac (main effect $p = 0.007$) and, marginally so (main effect $p = 0.092$), for the 2° Vac. For both the 1° and 2° Vac birds, IL6 expression was elevated ($p \leq 0.05$) at 6 h and reached higher levels in Alum&mIgG immunized birds (main effect $p = 0.007$ and $p = 0.034$ in 1° and 2° Vac birds, respectively). For IL8 (CXCL8) levels and duration of expression were similar with 1° and 2° Vac, reaching peak levels at 6 h, then dropped to lower levels but remained elevated ($p \leq 0.05$) at 1 to 3 d.

### 3.3.2. IL10 Expression

IL10 expression fluctuated near baseline levels (time main effect $p = 0.107$) in mIgG-antigen injected GF-pulps from unsensitized chickens throughout the 5-day time-course examined. In 1° Vac chickens, independent of immunization treatment, IL10 expression was elevated ($p \leq 0.05$) at 6 h to 2 d, with levels approximately 50-fold above 0 h levels. Levels then gradually decreased to pre-injection levels on 7 d. Overall, IL-10 expression was higher (main effect $p = 0.025$) in GF-pulps of Alum&mIgG compared to mIgG immunized chickens. IL10 expression in 2° Vac birds was more than 200-fold higher at 6 h than before mIgG injection (0 h), then dropped to lower but elevated levels on 1–3 d, before returning to near pre-injection levels at 4 and 5 d. There were no immunization treatment differences in IL10 levels in mIgG injected pulps from 2° Vac birds.

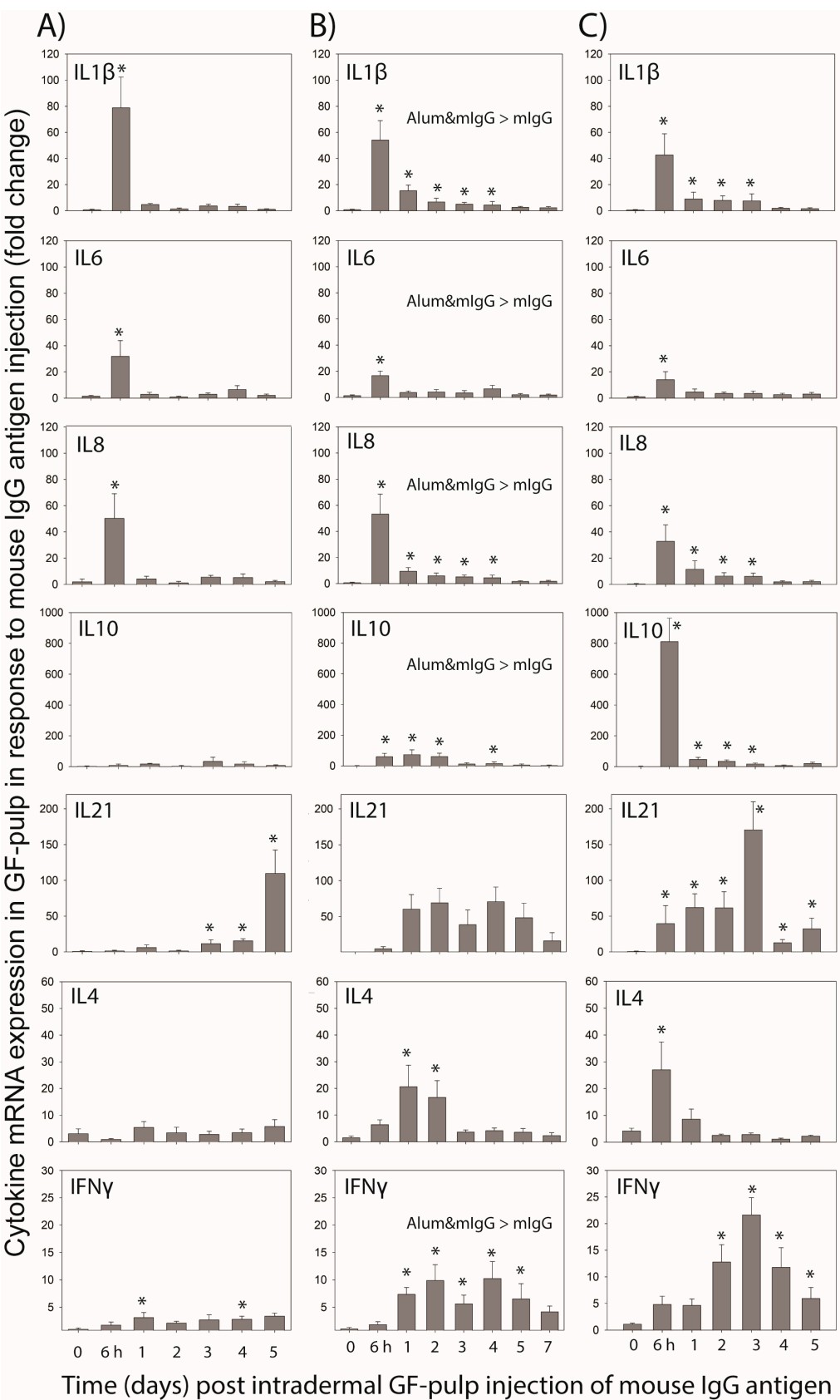

**Figure 3.** Relative mRNA expression of cytokine genes in response to intradermal injection of mouse IgG-antigen into the pulp of growing feathers (GF) in unsensitized (sham-(PBS) immunized) chickens (**A**), and in immunized chickens on Day 10 after the primary (**B**) or Day 5 after the secondary

(**C**) intramuscular immunization with mouse IgG in PBS (mIgG) or mixed with alum adjuvant (Alum&mIgG). For immunization, male Light-brown Leghorn chickens were injected i.m. with 26 µg of mIgG in endotoxin-free (EF) PBS or mIgG mixed with 15% alum adjuvant in EF-PBS; four chickens per immunization treatment. Another four chickens were i.m. injected with the same volume (0.1 mL) of EF-PBS vehicle. The 1° and 2° immunizations were conducted when the chickens were 7 and 11 weeks of age, respectively. To determine cellular responses to mIgG, the pulps of 20 GFs were injected i.d. with mIgG (1 µg/µL; 10 µL/GF) 10 d after the primary, or 5 d after the secondary i.m. immunization, or in age-matched sham (PBS) immunized chickens. Injected GF collected before (0) and at 0.25, 1, 2, 3, 4, and 5 d post GF-injection were used to conduct targeted gene-expression analysis by qRT-PCR (one GF per time-point and chicken). As no time by treatment interactions were observed, data shown are the time main effect means $\pm$ SEM (n = 8 chickens at each time-point) of fold change in cytokine gene expression compared to a calibrator sample included in each assay plate. * Within a treatment group, main effect mean-values at a time-point are higher ($p \leq 0.05$) than before i.d. pulp injection of mIgG antigen (0 d). When present, main treatment differences between mIgG and Alum&mIgG vaccination treatments are indicated within a graph ($p \leq 0.05$).

### 3.3.3. IL21 Expression

IL21 expression in mIgG-antigen injected GFs of unsensitized chickens started to increase on 3 d and continued to increase to approximately 50-fold higher expression on 5 d compared to 0 d. In 1° Vac birds, there was a marginal main effect of time for IL21 expression ($p = 0.05$), with elevated levels from 1–5 d, although, due to the high variation, IL21 levels were not different from 0 time at any of the time-points examined. In 2° Vac birds, IL21 expression was greatly elevated at 6 h, reached peak levels (more than 150-fold higher than 0 d) by 3 d, then dropped to lower, but still elevated levels at 4 and 5 d. For both 1° and 2° Vac chickens, there were no differences in IL21 expression due to vaccination treatment.

### 3.3.4. IL4 Expression

In mIgG-antigen injected GF from unsensitized chickens, there were no significant changes in IL4 expression over time. In 1° Vac birds, however, IL4 expression levels increased within 6 h reaching maximal levels on 1 and 2 d ($p \leq 0.05$), and then returned to near pre-injection levels 3–7 d. In 2° Vac birds, IL4 expression was only elevated ($p \leq 0.05$) at 6 h, dropped to near pre-injection levels on 1 d and remained at preinjection levels thereafter. For both 1° and 2° Vac chickens, there were no differences in IL4 expression due to vaccination treatment.

### 3.3.5. IFNγ Expression

Relative IFNγ mRNA levels were elevated ($p < 0.05$) early (1 d) in mIgG-antigen injected GF of unsensitized chickens and remained at this elevated level (approx. 5-fold above 0 h levels) throughout the 5-day examination period. In 1° Vac birds, IFNγ expression was elevated ($p \leq 0.05$) at 1 d and remained elevated near the 1 d level at 2 to 5 d. Overall, IFNγ expression was higher (treatment main effect $p = 0.041$) in Alum&mIgG compared to mIgG immunized birds. In 2° Vac birds, IFNγ levels increased at 6 h and were significantly elevated ($p \leq 0.05$) from 2 to 5 d, with peak levels on 3 d. There were no differences in IFNγ expression due to immunization treatments.

## 4. Discussion

Using the growing feather (GF) as a skin-test site [10] to monitor and evaluate cellular/tissue immune system responses in an individual, is a relatively novel tool, unique to the avian model. Because the living portion (pulp) of the GF consists of a column of dermis enveloped by epidermis, and GFs are loosely attached in the skin (unlike mature feathers), each GF can serve as a minimally invasive skin biopsy sample. Intra-dermal injection of test-material into multiple GFs in an individual and collection of injected GFs

at different time-points, therefore, enables monitoring of cellular/tissue responses taking place in vivo—similar to monitoring antibody responses in blood. The main objective of this study was to demonstrate the possibility of using the GF in-vivo test-tube system together with blood sampling to simultaneously monitor local cellular- and systemic antibody-responses to a protein antigen, specifically mIgG, in the same individuals. The observations reported here strongly support the feasibility of this approach and its effectiveness in providing windows into systemic and local immune system activities taking place in vivo in an individual.

For immunization, the T-dependent test-antigen, mouse IgG (mIgG) was administered i.m. as mIgG in PBS vehicle or mixed with alum adjuvant prepared in PBS (mIgG and Alum&mIgG, respectively). The humoral response was monitored for 28 d following a primary and a secondary i.m. immunization in immunized chickens. The same blood sampling protocol also was carried out in age-matched chickens that were injected i.m. with vehicle (PBS) (unsensitized chickens). mIgG and Alum&mIgG i.m. immunizations resulted in mIgG-specific antibody production. As expected, mIgG-specific IgM antibody levels were similar post-primary and -secondary immunization, whereas the IgG antibody profiles exhibited characteristics of primary and memory responses to a T-dependent antigen, respectively (i.e., isotype-switch to IgG, higher and faster increases in circulating antibody levels post-secondary immunization). Mouse IgG, a relatively large (~150 KD), xenogeneic, complex protein was able to induce a strong antibody response without the help of an adjuvant, although mixing mIgG with a known adjuvant, alum [18], resulted in stronger primary and memory IgG responses than mIgG alone.

Examination of the local tissue/cellular response to mIgG-antigen injected i.d. into GFs in unsensitized chickens and in immunized chickens during the primary- and secondary-response effector phases (10 d and 5 d post primary and secondary immunization, respectively) revealed novel information regarding leukocyte infiltration- and cytokine gene-expression (mRNA)-profiles at the antigen injection site. While there are no similar time-course data in the literature, our observations are in line with established trends of faster and more prominent leukocyte infiltration in immunized compared to unsensitized individuals, including heterophil (neutrophil) infiltration followed by monocytes/macrophages and lymphocytes [1,10]. Heterophils, the first responders during an inflammatory response, consistently reached peak-levels at 6 h following antigen injection into the dermis of GFs, independent of the type of immunization. Antigen injection into GFs during the secondary effector response resulted, however, in higher levels of heterophil infiltration in chickens immunized with Alum&mIgG. Similarly, macrophage infiltration peaked at 6 h in immunized birds, and was greater in birds immunized with Alum&mIgG, especially with the secondary immunization. Inclusion of the alum adjuvant likely stimulated inflammatory mechanisms that heightened the heterophils' and monocytes/macrophages' responsiveness to recruitment signals in response to re-introduction of the antigen in immunized chickens. Further studies are needed to better explain this observation.

All subsets of T cells examined (γδ-TCR+, CD4+, or CD8+) were found to infiltrate mIgG injected GFs, independent of immunization status and treatment. T cell infiltration occurred earlier and at higher levels in immunized chickens, especially during the secondary effector response. Together, these observations are indicative of a T cell memory-type response to mIgG antigen. With the exception of γδ T cells, where Alum&mIgG immunization resulted in higher γδ T cell recruitment than mIgG immunization, inclusion of adjuvants for immunization did not affect the T cell infiltration response. Moreover, γδ T cells reached their higher peak levels early, at 6 h and/or 1 d, in immunized birds, whereas in unsensitized birds, levels were elevated on 1 d and remained at this level through 5 d. Considering that γδ T cells do not require processing and presentation for activation via their T cell receptor, their early response and cytokine production ability may be important in directing the nature of the local response [1,10,22].

Surprisingly, levels of CD4+ T cells and CD8+ T cells, although higher at 6 h in immunized compared to unsensitized birds, were similar throughout the 5–7 d examination

period for all immunizations. Functionally, the CD4+- and CD8+-T cells likely are helper and cytotoxic T cells, respectively, although the CD8+ T cell population may include γδ T cells. Considering that the test-antigen, mIgG is a soluble protein antigen, responding T cells are expected to primarily play a supporting role (e.g., via IFN-γ production) rather than a direct role in the elimination of the antigen. Therefore, faster not necessarily greater recruitment of activated cytokine producing T cells to the affected tissue may be the most important benefit of prior immunizations. It is also possible that the proportions of antigen-specific cells among recruited T cells increased with repeat exposure, an aspect that was not examined in this study but is likely to be an important part of the improved T cell effector response in sensitized individuals [1].

B cell infiltration profiles exhibited the most striking differences between types of immune responses and between immunization treatments. B cell levels in mIgG-injected GFs reached maximal levels by 5 d, 3 d and 1–2 d in unsensitized chickens and chickens during the primary- and memory-effector responses, respectively. In immunized chickens, the B cell infiltration response was lower with Alum&mIgG than with mIgG. The function and specificity of the B cells recruited to the antigen-injected dermal tissue is not clear from this study. These B cells may be actively secreting antibodies, carrying out B cell receptor-mediated antigen removal, presenting antigens for the local activation of T cells, and/or producing cytokines and chemokines. We observed a similar B cell presence in GFs and other skin test-tissues (i.e., wattles and wing webs) injected with *Mycobacterium butyricum* (bacterin) in *M. butyricum* immunized chickens [10], as well as in wattles during the recall response to melanocyte-lysates in chickens with a Th1-mediated melanocyte-specific autoimmune disease [3]. Participation of B cells in the induction of cell-mediated responses also was reported for cell-mediated (DTH) responses in mice, especially in response to soluble protein antigen [23,24] and human and animal studies reported antigen activated B cell recruitment to the site of infection [25]. In humans with tuberculous pleurisy, pleural B-1 (IgM+) B cells were described to participate in the cell-mediated response to *M. tuberculosis*, where they were shown to exert a homeostatic effect by producing IL-10 [26,27]. Further research is needed to address the morphological and functional phenotype of B cells participating in the local cellular/tissue response to protein antigen in this chicken model.

The mIgG-injected GFs also were collected to conduct targeted gene-expression analysis at the transcriptome level for IL-1, IL-4, IL-6, IL-8, IL-10, IL-21, and IFN-γ cytokines. Expression of these cytokines and chemokine IL-8 (CXCL8) were previously shown to be initiated at different times and different levels during local innate and antigen-specific responses in chickens [3,13,14,17,28–31]. Considering the test-antigen (mIgG) used here is a soluble protein, without microbial components, it is not surprising that expression of pro-inflammatory cytokines IL-1, IL-6 and IL-8, although initiated early, was short-lived and expression profiles were similarly independent of whether mIgG was administered for the first time in unsensitized chickens or in mIgG- or Alum&mIgG-immunized chickens. It should be noted, however, that in unsensitized chickens, local expression profiles of IL-1β and IL-8 cytokines reached peak levels at 6 h and rapidly dropped to pre-injection levels at 1 d through 5 d; whereas in immunized chickens, their mRNA expression levels also peaked at 6 h but remained elevated for 3 to 4 days. Independent of immunization treatment, these inflammatory cytokines may be produced by local macrophages and other tissue cells in response to mIgG, as part of the innate response to the presence of a foreign protein and to the tissue injury caused by the injection [1,10,17]. The early expression of these inflammatory cytokines parallels the spike in heterophils, the recruitment of which is another strong indicator of local innate inflammatory activity, while their later and/or sustained expression likely supports macrophage recruitment and activation. Similar heterophil and macrophage infiltration profiles and concurrent expression of IL1β, IL6, and IL8 mRNA were observed when the acute inflammatory response to lipopolysaccharide was examined in broilers using the GF in vivo test-tube system [17]. Notable also is the higher expression of these cytokines in mIgG injected GF during the primary effector phase in chickens immu-

nized with Alum&mIgG compared to mIgG, attesting to the pro-inflammatory properties of the alum adjuvant.

Cytokines IL-21, IFN-γ, IL-4 and IL-10 are more strongly associated with T effector cell activities, although B cells and cells of innate immunity also are able to express these cytokine genes (e.g., IFN-γ by natural killer cells; IL-4 by mast cells; IL-21 by natural killer cells; IL-10 by B cells; and macrophages) [1]. The earlier and higher expression levels of these cytokines in GFs injected with mIgG during the primary- and secondary-effector responses in immunized chickens compared to the response observed during a first exposure to mIgG in unsensitized chickens, supports that activated lymphocytes, especially T cells, of adaptive immunity are the likely source of these cytokines. Additionally, when comparing the time-course of cytokines IFN-γ and IL-4 known to suppress each other's expression, the inverse relationship between these cytokines becomes evident particularly during the secondary-effector response, suggesting T helper cell-type 1 polarization and memory phenotype of the cellular response. The earlier and higher expression of these cytokines, from the response in unsensitized chickens to the primary and secondary effector responses in immunized chickens, is consistently observed in this study. Particularly, the time-course of IL-21 expression differed greatly in antigen-injected GFs of unsensitized and immunized chickens; i.e., a steady increase over the 5-d study versus a rapid (by 6 h) and sustained elevation in expression, respectively. The IL-21 expression profiles parallel the B cell infiltration in unsensitized chickens (0–5 d), and the early phase of B cell infiltration in immunized chickens. While IL-21 is known to be secreted by follicular T helper cells to support B cell differentiation, the association between this cytokine and B cell participation in the local response to antigen needs to be further examined. Expression profiles of IL10 differed most dramatically between the types of immune responses examined, with no significant IL10 mRNA expression in unsensitized chickens, to greatly increased and sustained expression levels with immunization, especially when antigen was injected into GF during the secondary effector phase. IL10 expression during the primary effector response also differed between immunization treatments, with higher IL10 expression observed in antigen-injected GF of chickens immunized with Alum&mIgG. Hence, alum adjuvant also seems to support anti-inflammatory activities in the initiation of the adaptive immune response to mIgG-antigen. The source of IL10 expression in antigen-injected GFs likely involves T cells and B cells recruited to the site of antigen injection in addition to activated macrophages. The relative contribution of these cells to local IL10 expression needs to be further examined.

## 5. Conclusions

This study is a first to simultaneously evaluate systemic and local immune system responses to protein antigen over time in the same individuals. The response profiles generated revealed new insight into temporal, qualitative, and quantitative aspects of both cell- and antibody-mediated adaptive immune responses to protein antigen. Most notable was the memory phenotype observed in both the systemic humoral (plasma IgG antibody profiles) and the local cellular responses (leukocyte- and cytokine expression-profiles in GF-pulps) in chickens receiving a second immunization with mIgG or Alum&mIgG. Moreover, effects of adjuvant inclusion with antigen on both humoral and local cellular immune responses were revealed. The new knowledge gained from this study not only supports the unique opportunity in the avian system to monitor both systemic and local, innate-, primary- and memory-immune system activities taking place in vivo, but also provides direction for comprehensive scientific investigations on immune system function in poultry. The minimally invasive, two-window approach used here likely will find direct application in poultry breeding and management strategies designed to optimize poultry health and well-being.

**Author Contributions:** Conceptualization, G.F.E. and K.A.B.; animal experiments all authors; laboratory procedures and data acquisition H.R.K. (ELISA and qRT-PCR), D.M.F. and K.A.B. (IF- staining and flow cytometry); data analyses G.F.E. and D.M.F.; writing, original draft G.F.E.; review and

editing by all authors; funding acquisition, project administration, supervision G.F.E. All authors have read and agreed to the published version of the manuscript.

**Funding:** Research reported in this publication was supported in part by the National Institute of Biomedical Imaging and Bioengineering of the National Institutes of Health under grant number R15 EB 015187-01A1; GF Erf, PI; and the Tyson Endowed Professorship in Avian Immunology; GF Erf.

**Institutional Review Board Statement:** All procedures were approved by the University of Arkansas Institutional Animal Care and Use Committee (Protocol #15020) before the study was initiated.

**Data Availability Statement:** Data are available upon request from the corresponding author.

**Acknowledgments:** The authors appreciate the access to the flow cytometer and the real-time PCR instrument in the University of Arkansas System Division of Agriculture's Cell-Isolation and Characterization Facility and Central Molecular Laboratory, respectively, housed in the Center of Excellence for Poultry Science, Fayetteville, Arkansas.

**Conflicts of Interest:** The authors declare that the research was conducted without any commercial or financial relationships that could be construed as a potential conflict of interest. The funders had no role in the design of the study; in the collection, analyses, or interpretation of data; in the writing of the manuscript; or in the decision to publish the results.

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
