# Peer review of "Two-Window Approach to Monitor and Assess Cellular and Humoral Immune Responses in Poultry"

_poultry, doi:10.3390/poultry2010009_

Round 1

Reviewer 1 Report

This paper by Erf et al. builds on their previous work describing a new technology to investigate immune responses in a “test tube” the growing feather (GF). While quantification of humoral immune responses straight forward by using plasma samples to quantify immunoglobulin levels, analyzing cell-mediated immune responses has been very difficult, cumbersome and highly variable. The Erf group developed a new system to overcome this problem using the growing feather to inject immune modulators and antigens. Here they build on their previous work to investigate the secondary humoral and cellular response after primary intradermal injection of a model antigen (murine IgG) in the GF system. They show that both the immunoglobulin response as well as cellular responses quantified by cytokine expression and leucocyte infiltration can be easily measured in comparison to standard technologies of skin testing.

The study is well performed and data are clearly presented. Importantly, this work stimulates a host of questions when reading the paper, the work lays the ground to address them in the future. Thus, I strongly recommend the paper for publication in “Poultry” after a few minor comments are addressed.

Line 114: please provide metric measures in addition.

Line 191: this should read delta, delta CT (ΔΔCt)

Line 281: I find this sentence difficult to understand. GF-pulps were injected 10 and 5 days post primary and secondary intradermal injection respectively, as I understand. Please describe it in more detail in the figure legend (even though this might be redundant).

Line 270: could you provide absolute numbers recovered from the GF? This might be helpful for those trying to use the method in the future.

Line 469; please indicate that the CD8+ population does include γδ T-cells and not only CTLs (even though some γδ T-cells may have cytotoxic properties

Lien 528: could you discuss the cytokine response to the T-cell depended antigen in comparison to LPS injection induced response quantified in your earlier work?

Author Response

Response to Reviewer 1

Thank you for your review and helpful comments. As you describe it, our major goal to demonstrate the opportunities in poultry immunology to study, monitor, and assess cellular responses taking place in vivo at the site of test-material injection in naïve and immunized chickens.  Here, we specifically examined systemic humoral and local tissue cellular responses to a protein antigen (mouse IgG) in the same individual. Indeed, in addition to demonstrating the suitability and application of this minimally invasive approach, we are reporting new knowledge and stimulating a host of questions, mainly because such longitudinal studies on local tissue immune responses were not previously conducted.

Authors response/corrections:

Line 114: 0.5 inch = 12.5 mm

Line 191: delta delta – corrected

Line 281: Figure caption was extended to clarify that GF pulps were injected with mIgG 10 and 5 days post-primary and secondary intramuscular immunization, respectively with mIgG or Alum&mIgG. This was done for Figure 2 and 3.

Line 270: re. reporting cell numbers isolated from GF-pulp.  We could not find the exact place the reviewer is referring to but agree this will be helpful information when seeking to isolate cells for other analyses.  Our approach is generally to express data as % total pulp cells, as the pulp unit is equal at time of injection, all cell suspensions are brought to the same volume, and the same volume of cell suspension is used for immunostaining and cell population analysis by flow cytometry.  Naturally with the recruitment and infiltration of leukocytes the numbers of cells in the “pulp unit” changes.  We found that expressing leukocyte presence as a percentage of total pulp, is an informative and “ relatively quantitative” estimate reflecting local activities.      

Line 469: statement that the CD8+ population does include CD8+ γδ T cells was added.

Line 528: a statement comparing the cytokine response to the T dependent antigen with that to LPS was added.

Reviewer 2 Report

The present research was conducted to demonstrate the application of the two-window approach to simultaneously examine and assess adaptive cellular and humoral immune responses, over-time, in an individual , and demonstrate the suitability of this two-window approach to monitor and assess vaccine responses .  I think that the manuscript is convenient with the scope of the journal . The paper could provide information of interest in this field because it demonstrates the unique opportunity in the avian model to monitor both cell- and antibody-mediated immune responses using minimally invasive techniques

Author Response

Authors: 

Thank you for your review of this manuscript and for confirming that the studies reported demonstrate the suitability and opportunities of the two-window approach to monitor and access both cell- and antibody-mediated immune responses using minimally invasive techniques.

Reviewer 3 Report

Manuscript No.poultry-2195301 entitled'' Two-window approach to monitor and assess cellular and humoral immune responses in poultry''. This manuscript has demonstrated the suitability of this two-window approach to monitor and assess vaccine responses. I have some concerns; thus, I suggest the authors address those concerns in the revised version.

1- The abstract is general, please describe your most important results and add a p-value for each result and make the conclusion clearer according to the major obtained data.

2- In the introduction, a lot of statements without the corresponding citation, please revise the citations of all statements.

3- In the statistical analysis section data should be subjected to normality and homogeneity tests before analysis.

4- Please describe all abbreviations in their first mention, it is preferable to add a list of abbreviations 

5- Please proofread the whole manuscript to avoid grammatical errors.

6- Too long conclusion section, please be focused on the most important results.

Author Response

Reviewer 3.

Authors: Thank you for review of this manuscript and for agreeing that we demonstrated the suitability of this two-window approach to monitor and assess cellular and humoral immune responses. 

Your comments are much appreciated. 

Reviewer: Manuscript No.poultry-2195301 entitled'' Two-window approach to monitor and assess cellular and humoral immune responses in poultry''. This manuscript has demonstrated the suitability of this two-window approach to monitor and assess vaccine responses. I have some concerns; thus, I suggest the authors address those concerns in the revised version.

Reviewer 1- The abstract is general, please describe your most important results and add a p-value for each result and make the conclusion clearer according to the major obtained data.

Authors 1- With the word limit for the abstract and the provided guidelines to prepare an abstract that summarize the basis, purpose, experimental approach, results and conclusions, we feel that the abstract as written meets these guidelines and the major findings are correctly reported. After trying several revisions, it seems impossible to add P-values without individual, lengthy descriptions of the many findings from this complex study.

Reviewer 2- In the introduction, a lot of statements without the corresponding citation, please revise the citations of all statements.

Authors 2- The introduction initially mentions basic immunology concepts, hence the textbook citations. The use of the skin as a test-site mentions 10 citations, then the background on the GF as a cutaneous test-site provides another 6 citations. This is followed by citation for the dual window approach and the use of Alum adjuvant. As is, we feel, the statements in the introduction are properly and fully cited.

Reviewer 3- In the statistical analysis section data should be subjected to normality and homogeneity tests before analysis.

Authors 3- The SigmaPlot software automatically includes normality (Shapiro-Wilk) and equal variance tests (Brown-Forsythe) for ANOVA and Repeated Measures ANOVA and provides guidelines for alternative analyses should tests fail.   

Reviewer 4- Please describe all abbreviations in their first mention, it is preferable to add a list of abbreviations 

Author 4- We high-lighted abbreviations when first used.

Reviewer 5- Please proofread the whole manuscript to avoid grammatical errors.

Authors 5- Done

Reviewer 6- Too long conclusion section, please be focused on the most important results.

Authors 6 - We feel the conclusion is focused on the most important results. Most of the GF data are new knowledge, as is the concurrent analysis of the humoral and cellular responses. We thought about deleting the last statement regarding applications of this minimally invasive approach. However, considering the nature of the manuscript as a "featured paper " for Poultry“, it is important to include it.

Reviewer 4 Report

Dear authors

The manuscript has some critical errors. Please check the following comments. Its confusing why two-way analysis was performed (line 205)? And how? Because the samplings were done a period of time the statistical analysis has to be changed to “repeated measure”. The figures and tables have to be self-standing. Then please define and explain all abbreviations including “PBS” etc. The figures have to include superscript indicators rather than asterisk. As a reader, I can't understand which treatment or time is higher than another one. Please revise.

The introduction to your manuscript provides a comprehensive overview of the background and significance of your study, as well as the main objective and methods. Here are a few suggestions for improvement:

Literature review: Discuss previous studies on monitoring and assessing immune responses in poultry and how they have advanced our understanding in this field. Mention how the current study builds upon or differs from previous studies.

Importance of the two-window approach: Please explain why using a two-window approach (GF pulp injection and blood sampling) is a significant advancement in the field and why it is necessary to simultaneously examine both cellular and humoral immune responses in poultry.

Advantages of GF pulp injection: Please discuss the unique benefits of GF pulp injection as a tool for monitoring cellular immune responses in poultry. Explain why it provides a minimally invasive, yet informative, method for studying local tissue responses in an individual.

Significance of mIgG antigen: please discuss the importance of using mouse IgG as a test antigen in poultry. Explain why mIgG is a T-dependent antigen and why it is known to stimulate antibody production in chickens.

Importance of Alum adjuvant: please discuss the role of Alum in stimulating both cellular and humoral immune responses in poultry and how its presence in the mIgG antigen may impact the immune response.

You may also consider adding relevant references to support your claims and provide additional context to your reader. These references can come from academic journals in the fields of veterinary science, immunology, or poultry science, and should be appropriately cited in the text.

Author Response

Reviewer 4.

Thank you for reviewing the manuscript and your comments. We will address them individually below.

We do feel however, that many of the comments were addressed in the manuscript and some of the suggestions are outside the scope of the purpose of the study and the publication.

  1. Reviewer: The manuscript has some critical errors. Please check the following comments. Its confusing why two-way analysis was performed (line 205)? And how? Because the samplings were done a period of time the statistical analysis has to be changed to “repeated measure”. The figures and tables have to be self-standing. Then please define and explain all abbreviations including “PBS” etc. The figures have to include superscript indicators rather than asterisk. As a reader, I can't understand which treatment or time is higher than another one. Please revise.

Author 1: Sorry for the confusion, but overall the analyses and descriptions are correct and precise. 

Line 205: one-way analysis was performed for data from unimmunized (PBS immunized) chickens injected i.d. with mIgG-antigen in GF.  There is only one treatment, hence only the time effect was examined.

For immunized chickens, 2-way ANOVA were conduced to test the effect of time, immunization treatment, and their interaction.  As there were no interactions, time main effect means across treatments are shown.  Treat immunization effects (Alum&mIgG vs mIgG) are indicated at time points when they were present. Since only two immunization treatments were examined indication of a difference should be clear.

We agree, for blood sampling, repeated measures ANOVA was appropriate as we are repeatedly collecting from the same pool of blood in an individual.  However, each GF is its own unit.

We chose to only indicate time-differences in comparison to pre-injection levels, as the overall time-course profiles are very visual, and indication of individual time-comparison makes the figure too busy. As is, the Figures are showing critical statistical finding that together with the Result narrative, figures, and figure captions, are informative regarding changes/differences in the various aspects examined.

  1. Reviewer: The introduction to your manuscript provides a comprehensive overview of the background and significance of your study, as well as the main objective and methods. Here are a few suggestions for improvement:

Authors 2. Thank you for your excellent suggestions for improvement!  We agree that these are interesting points to include, especially if it were a comprehensive literature review for a thesis/dissertation or a review paper on the subject matter.  However, for this introduction we provided the necessary background and significance for this study as expected for a research paper.